# MODELING EMERGENT LEXICON FORMATION WITH A SELF-REINFORCING STOCHASTIC PROCESS

**Brendon Boldt, David Mortensen**
Language Technologies Institute
Carnegie Mellon University
Pittsburgh, PA 15213, USA
`{bboldt,dmortens}@cs.cmu.edu`

## ABSTRACT

We introduce FILEX, a self-reinforcing stochastic process which models finite lexicons in emergent language experiments. The central property of FILEX is that it is a *self-reinforcing* process, parallel to the intuition that the more a word is used in a language, the more its use will continue. As a theoretical model, FILEX serves as a way to both explain and predict the behavior of the emergent language system. We empirically test FILEX's ability to capture the relationship between the emergent language's hyperparameters and the lexicon's Shannon entropy.

## 1 INTRODUCTION

The methods of emergent language provide a uniquely powerful way to study the foundational questions of how language arises and develops. While many papers have used emergent language to empirically study such properties, there has been a relative lack of theoretical models accounting for the empirical data. Such models are important insofar as they can both *explain* and make *predictions* about the behavior of the actual system.

To this end, we introduce FILEX, a self-reinforcing stochastic process based on the Chinese restaurant process, as a model of the Shannon entropy of an emergent language's lexicon. This simple process models a comparatively complex emergent language system[1] (ELS), comprising environment dynamics, neural networks, gradient descent, and reinforcement learning. While the complex system cannot be reasoned about directly, FILEX is simple enough for this. In Section 2, we explain the emergent language system which we are modeling. Section 3 introduces FILEX and how it corresponds to presented ELS. Finally, we empirically test FILEX's ability to capture the relationship between the emergent language's hyperparameters and the lexicon's Shannon entropy in Section 4.[2]

**Related Work**  For a survey of deep learning-based emergent language work, please see Lazaridou & Baroni (2020). The environment of this paper is most analogous to the near-continuous color-naming environment of Chaabouni et al. (2021) because both environments deal with the discretization of a (near) continuous signal into a smaller number of atomic, discrete symbols. Insofar as this paper studies the entropy of emergent language, it works in a similar space to Kharitonov et al. (2020); Chaabouni et al. (2021). Mordatch & Abbeel (2018) use the Chinese restaurant process as an auxiliary reward to inductively bias the emergent language learning process. Although this partly inspired the model presented in this work, the methodological roles of an auxiliary reward and theoretical model are distinct.

On a methodological level, this paper is similar to Resnick et al. (2020) which also develops a theoretical model for an emergent language phenomenon. Namely, their model predicts the compositionality of an emergent communication protocol dependent on the capacity of the neural-network agents. Analogously, out model predicts the entropy of the emergent communication protocol dependent on learning rate, replay buffer size, bottleneck size, or training steps.

---

[1]*Emergent language system* or ELS refers to the combination of agents (neural networks), the environment, and the training procedure used as part of an emergent language experiment.

[2]Code is available at
`https://github.com/brendon-boldt/simple-emergent-navigation`

## 2 EMERGENT LANGUAGE SYSTEM

**Environment**    In this paper we use a simple 2-dimensional, obstacle-free navigation environment. A sender agent observes the position of a receiver agent, sends a message to the receiver, and the receiver takes a step. For a given episode, the receiver is initialized uniformly at random within a circle and must navigate towards a smaller circular goal region at the center. An illustration is provided in Appendix A. The receiver's location and action are continuous variables.

**Agent Architecture**    Our architecture comprises two agents, conceptually speaking, but in practice, they are a single neural network. The *sender* is a disembodied agent which observes the location of the receiver and passes a message in order to guide it towards the goal. The *receiver* is an agent which receives the message as its only input and takes a step solely based on that message. The sender and receiver are randomly initialized at the start of training, trained together, and tested together.

The observation of the sender is a pair of floating-point values representing the receiver's location. The sender itself is a 2-layer perceptron with tanh activations. The output of the second layer is passed to a Gumbel-Softmax bottleneck layer (Maddison et al., 2017; Jang et al., 2017) which enables learning a discrete, one-hot representation.[3] The activations of this layer can be thought of as the words forming the lexicon of the emergent language. At evaluation time, the bottleneck layer functions deterministically as an argmax layer, emitting one-hot vectors. The receiver is a 1-layer perceptron which takes the output of the Gumbel-Softmax layer as input. The output is a pair of floating-point values which determine the step direction and magnitude of the receiver. An illustration and precise specification are provided in Appendices A and B.

**Optimization**    Since our environment involves multi-step rewards, we use proximal policy optimization (PPO) (Schulman et al., 2017) paired with Adam to optimize the neural networks; we use PPO specifically because it is more stable than a vanilla policy gradient method. We the PPO implementation of Stable Baselines 3 built on PyTorch (Raffin et al., 2019; Paszke et al., 2019).

**Rewards**    We make use of two different rewards in our configuration, a *base* reward and an *shaped* reward. The base reward is simply a positive reward of $1$ given if the receiver reaches to the goal region before the episode ends and no reward otherwise. The shaped reward, given at every timestep, is the decrease in distance to the goal, that is $|s| \cdot \cos \theta$, where $|s|$ is the size of the step, and $\theta$ is the bearing to the goal; for example, taking the maximum step size directly towards/away from the goal yields a reward of $1/-1$.

## 3 FILEX STOCHASTIC PROCESS

FILEX[4] is a mathematical model based on the Chinese restaurant process (CRP) (Blei, 2007; Aldous, 1985), an iterative stochastic process which yields a probability distribution over the positive integers. The basic idea of the CRP is that in a hypothetical restaurant, each table corresponds to a positive integer; as each customer walks in, they sit at a random table with a probability proportional to the number of people already at that table. The key property here is that the process is *self-reinforcing*; tables with many people are likely to get even more. By analogy to language, the more a word is used the more likely it is to continue to be used. For example, speakers may develop a cognitive preference for it, or it gets passed along to subsequent generations as a higher rate (Francis et al., 2021). Furthermore, self-reinforcing processes based on the CRP have been used to model the power-law distributions found in natural language (Goldwater et al., 2011).

**Formulation**    The pseudocode describing FILEX is given in Figure 1a. The process starts with an array of weights, length $S$, initialized to $\alpha/S$. $\alpha$ is a positive, real-valued hyperparameter controlling the uniformity of the final result (high $\alpha$, high uniformity). At each iteration, we select $\beta$ samples (i.e., indices) with replacement from a categorical distribution parameterized by the weights. For

---

[3]Using a Gumbel-Softmax bottleneck layer allows for end-to-end backpropagation, making optimization faster and more consistent than using a backpropagation-free method like REINFORCE (Kharitonov et al., 2020; Williams, 1992).

[4]Short for "finite lexicon stochastic process"

```
1    alpha: float > 0
2    beta: int > 0
3    N: int >= 0
4    S: int > 0
5
6    weights = array(size = S)
7    weights.fill(alpha / S)
8    for _ in range(N):
9      w_copy = weights.copy()
10     for _ in range(beta):
11       i = sample_categorical(w_copy)
12       weights[i] += 1 / beta
13   return weights / sum(weights)
```

(a) FILEX written in Python pseudocode.

| Ind. var. | ELS | FILEX |
|---|---|---|
| $1/\alpha$ | $-0.86$ | $-0.87$ |
| $\beta$ | $+0.84$ | $+0.95$ |
| $N$ | $-0.61$ | $-0.53$ |
| $S$ | $+0.65$ | $+0.77$ |

(b) Kendall rank correlation coefficient for each hyperparameter in both the ELS and FILEX. All values have $p < 0.005$

Figure 1

each, index, we increment the weight at the index by $1/\beta$ so the total update for every iteration is always 1. This proceeds $N$ times after which the weights are normalized to 1 and returned.

The two key differences between FILEX and the CRP are the hyperparameters $S$ and $\beta$. FILEX only has finitely many parameters so as to match the fact that the agents in the ELS have a fixed-size bottleneck layer, that is, a fixed, finite lexicon. Secondly, $\beta$ is introduced to account for the fact that certain RL algorithms like PPO accumulate a buffer of steps from the environment with the same parameters before performing gradient descent.

**Analogy** Identifying the correspondence between the emergent language system and FILEX is key to its explanatory value. Firstly, the weights of FILEX correspond the learned likelihood with which a given bottleneck unit is used in the ELS; in turn, both of these correspond to the frequency with which a word is used in a language. Each iteration of FILEX is analogous to a whole cycle in the ELS of simulating episodes in the environment, receiving the rewards, and performing gradient descent with respect to the rewards. In light of this, we can specifically draw connections between the four hyperparameters of FILEX. As mentioned before $\beta$ corresponds directly to the number of steps in the environment the agent takes before performing an update of the parameters. $S$ corresponds the size of the bottleneck layer in the ELS. $N$ corresponds the number of steps taken in the environment throughout the course of training the ELS.

$\alpha$ controls the initialization of weights of FILEX, and it corresponds, in part, to the *inverse* of the size of the parameter updates in the ELS, that is, the learning rate. We can see this by looking at the sum of the weights in FILEX: the sum is equal to $\alpha + n$ after the $n^{\text{th}}$ iteration. Since the both the categorical distribution and the final result normalize the weights, they are invariant to scaling by a positive number. Hence, we could scale by $1/\alpha$ and say that the initial weights sum to 1 with the sum being $1 + n/\alpha$ after $n$ iterations. Thus, varying $\alpha$ is equivalent to scaling the parameter updates by $1/\alpha$.

**Role of the Model** The key feature of a theoretical model is that it allows us to reason directly about how the phenomena in question work in order to direct subsequent empirical research. To achieve this, the model must make simplifications from the full system while still accurately describing some aspect of that system; in FILEX's case, we remove the environment and reward dynamics from consideration while specifically aiming to explain trends in lexicon frequency. On the other hand, the full, simulated system is necessary to verify that the behavior described by the model is, in fact, accurate. Wherever the model is inaccurate, it must either be revised or have its scope limited so as to remain in accord with the real system. In this way, the theoretical model and the ELS work in tandem to better the understanding of various phenomena in emergent language.

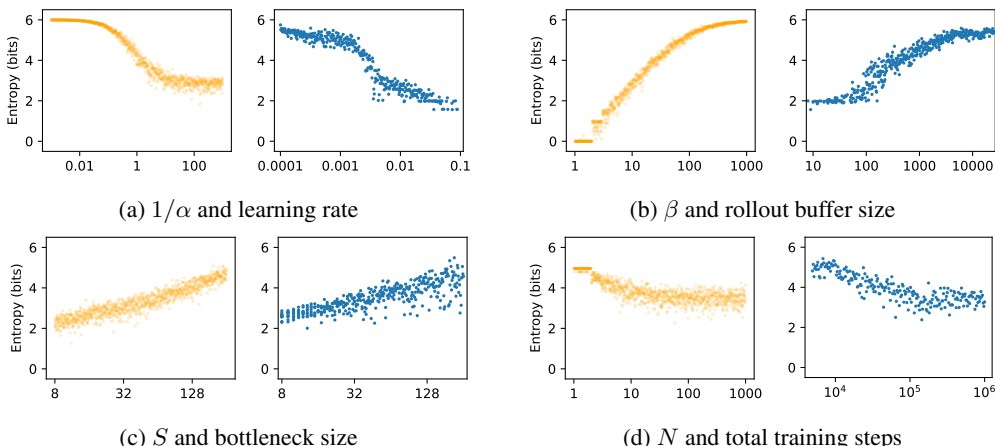

(a) $1/\alpha$ and learning rate

(b) $\beta$ and rollout buffer size

(c) $S$ and bottleneck size

(d) $N$ and total training steps

Figure 2: Plots for entropy vs. various hyperparameters. FILEX is plotted in orange and the ELS in blue.

## 4 EMPIRICAL EVALUATION

Our evaluations use the Shannon entropy (Shannon, 1948) of the lexicon as the dependent variable. For the ELS, the entropy is calculated from how frequently each of the bottleneck units are used across 3000 episodes; for FILEX, the entropy is calculated directly over the returned weights. Hyperparameter details are given in Appendix B. The quantitative evaluation is performed by comparing the Kendall rank correlation coefficient between each hyperparameter and entropy for both FILEX and the ELS, shown in Figure 1b. We see that FILEX both produces the correct rank correlation as well as approximating the degree of correlation.

Beyond measuring the correlation, we present the plots of entropy vs. the hyperparameters for both FILEX and the ELS in Figure 4 for qualitative evaluation. With the exception of the experiments varying $S$, $S = 2^6 = 64$, meaning that the maximum entropy is 6 bits. Although there is a conceptual correspondence between the hyperparameters of FILEX and ELS, the simplifications of the theoretical model mean that the numerical values of the hyperparameters are not in one-to-one correspondence with the ELS. In light of this, we choose hyperparameters for which FILEX best matches the behavior of the ELS; as a result, the $x$-axes of the plots are not identical.

**$\alpha$ and Learning Rate** FILEX's primary prediction regarding $\alpha$ is that as $\alpha$ increases, the entropy of the final distribution increases because for a high $\alpha$, the updates do not move the distribution far from uniformity whereas with a low $\alpha$, the updates make more a difference and can concentrate the distribution, leading to a lower entropy. This correlation (with $1/\alpha$) is borne out in the ELS experiment varying learning rate; additionally, both plots how a decreasing sigmoid pattern starting from the maximum entropy and going to a non-minimal entropy. Although the adaptive nature of the Adam optimizer weakens the analogy between these two hyperparameter, their correspondence is still evident empirically.

**$\beta$ and Rollout Buffer** FILEX's primary prediction regarding $\beta$ is that as $\beta$ increases, the entropy of the distribution increases because the updates to the weights at each iteration are "smoother," lessening the accumulation on particular weights. With one exception, this trend is matched by ELS, including the asymptotic shape as $\beta$/buffer size increases. The ELS plot, though, has a minimum entropy of 2 bits instead of 0 bits because the navigation requires 3 or more "words" (each corresponding to one direction) to span a 2-dimensional space[5], whereas no such restriction exists in FILEX.

**$S$ and Bottleneck Size** Note that in this experiment, $\alpha$ is increased in proportion to $S$ so that each individual weight has the same initialization for every $S$, otherwise, a constant $\alpha$ would be "spread

---

[5]2 bits of entropy corresponds to 4 actions; we do not currently have explanation for why the ELS never settles on 3, the theoretical minimum.

thin." FILEX's primary prediction regarding $S$ is that entropy will simply increase in proportion to $S$ which is matched by ELS experiment.

**$N$ and Training Steps**   FILEX's primary prediction regarding $N$ is that for low values, the entropy is maximized as the weights are close to their uniform initialization, but, as $N$ increases, the entropy decreases asymptotically to some non-minimal value. These predictions are generally matched by the ELS experiment.

## 5   CONCLUSION

In this paper we have introduced FILEX, based on the Chinese restaurant process, as a way to explain and predict the behavior of lexicon entropy in a simple emergent language system. FILEX is a model simple enough to reason about yet nuanced enough to capture the behavior the more complex system as borne out by experimentation. Building such theoretical models, generally speaking, is an important step in emergent language research as it moves the field toward developing general scientific accounts of the phenomena observed through empirical investigation. These scientific accounts are what will prove most useful in applying emergent language to linguistics and beyond.

ACKNOWLEDGEMENTS

This material is based on research sponsored in part by the Air Force Research Laboratory under agreement number FA8750-19-2-0200. The U.S. Government is authorized to reproduce and distribute reprints for Governmental purposes notwithstanding any copyright notation thereon. The views and conclusions contained herein are those of the authors and should not be interpreted as necessarily representing the official policies or endorsements, either expressed or implied, of the Air Force Research Laboratory or the U.S. Government.

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

## A    ENVIRONMENT AND AGENT ARCHITECTURE

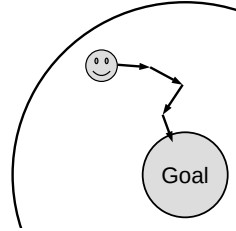

(a) The receiver (pictured) is rewarded for moving towards the goal region at the center.

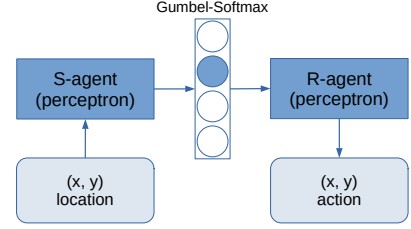

(b) The agent architecture is a single neural network although conceptually two separate agents: the sender and the receiver.

Figure 3

## B  HYPERPARAMETERS FOR EXPERIMENTS

### B.1  ELS EXPERIMENT CONFIGURATIONS

**Environment**   The environment is a circle 9 units wide with a circular goal region at the center which has a radius of 1 unit. The receiver can move up to 1 unit per step. The episode terminates after the $27^{\text{th}}$ if the agent has not reached the goal.

**Agent Architecture**

- Bottleneck size: $64$

- Architecture; sender is 1-3 and receiver is 5; bottleneck size is $N$

    1. Linear w/ bias: 2 in, 32 out
    2. Tanh activation
    3. Linear w/ bias: 32 in, $N$ out
    4. Gumbel-Softmax: $N$ in, $N$ out
    5. Linear w/ bias: $N$ in, 2 out (action) and 1 out (value for PPO)

- Bottleneck (Gumbel-Softmax) temperature: $1.5$

- Weight initialization: $\mathcal{U}\left(-\sqrt{\frac{1}{n}}, \sqrt{\frac{1}{n}}\right)$, where $n$ is the input size of the layer (PyTorch 1.10 default)

**Optimization**   Unless otherwise noted, we use the default hyperparameters for PPO as specified at `https://stable-baselines3.readthedocs.io/en/v1.0/modules/ppo.html`. Hyperparameters in Table 1 were changed from their default.

| Hyperparameter | Value |
|---|---:|
| Total training steps | $5 \times 10^4$ |
| Evaluation episodes | $3 \times 10^3$ |
| Learning rate | $3 \times 10^{-3}$ |
| Rollout buffer size | 256 |
| Batch size | 64 |
| Temporal discount factor ($\gamma$) | 0.9 |

Table 1: Optimization hyperparameters changed from their default value.

Learning rate corresponds applies to the whole end-to-end network.

**Experiments**   Each experiment uses a logarithmic sweep across hyperparameters; the sweep is defined by Equation 1, where $x$ and $y$ are the inclusive upper and lower bounds respectively and $n$ is the number steps to divide the interval into. The floor function is applied if the elements must be integers.

$$\text{LS}(x, y, n) = \left\{ x \cdot \left(\frac{y}{x}\right)^{\frac{i}{n-1}} \,\middle|\, i \in \{0, 1, \ldots, n-1\} \right\} \tag{1}$$

Table 2 provides the varied hyperparameters for the ELS experiments.

| ELS | Low | High | Steps |
|---|---:|---:|---:|
| learning rate | $10^{-4}$ | $10^{-1}$ | 200 |
| buffer size | $2^3$ | $2^{15}$ | 600 |
| bottleneck size | $2^3$ | $2^8$ | 400 |
| training steps | $10^2$ | $10^6$ | 400 |

Table 2: Logarithmic sweeps for each of the ELS hyperparameters. All other hyperparameters keep their default values.

## B.2 FILEX EXPERIMENT CONFIGURATIONS

The hyperparameters for the FILEX experiments are provided in Table 3.

| $\alpha$ | $\beta$ | $S$ | $N$ |
|---|---|---|---|
| $\mathrm{LS}(10^{-4}, 10^{-1}, 200)$ | $10$ | $64$ | $10^3$ |
| $10^{-3}$ | $\mathrm{LS}(2^3, 2^{15}, 600)$ | $64$ | $10^4$ |
| $5 \times 10^{-3} \cdot S^{\dagger}$ | $10$ | $\mathrm{LS}(2^3, 2^8, 400)$ | $10^3$ |
| $1$ | $5$ | $64$ | $\mathrm{LS}(10^2, 10^6, 400)$ |

Table 3: Hyperparameters for the empirical evaluation of FILEX. Each row corresponds to one experiment (which has one varying hyperparameter, specified by $\mathrm{LS}(\cdot, \cdot, \cdot)$ from Equation 1). $^{\dagger}\alpha$ is multiplied by the hyperparameter $S$ so that the individual weights are initialized to $5 \times 10^{-3}$.

## C  FURTHER EXPERIMENTAL RESULTS

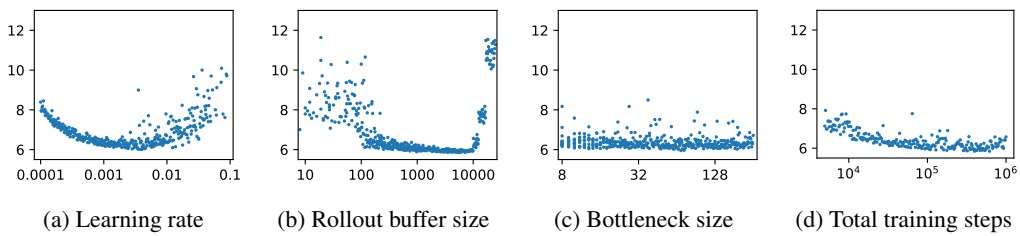

(a) Learning rate    (b) Rollout buffer size    (c) Bottleneck size    (d) Total training steps

Figure 4: Plots for average steps to complete episodes (lower is better) vs. various hyperparameters for the ELS. All data points have a $100\%$ success rate.

