# OpenReview forum: "Modeling Emergent Lexicon Formation with a Self-Reinforcing Stochastic Process"
_ICLR.cc/2022/Workshop/EmeCom — EmeCom Workshop at ICLR 2022_

### Official Review · Reviewer_8FGq · 2022-03-22
**Clear and nice theoretical approach.**

**Rating:** Accept
**Confidence:** 4

**Review:**

**Summary**

The goal of the paper is to propose a theoretical model, FiLex, that manages to explain and predict the behavior of emergent communication systems. Authors explain how their model captures the use of the lexicon by showing that their model help to predict lexicon’s entropy. Their model is an extension of the Chinese Restaurant Process with model parameters that can be linked to H-parameters of emergent communication experiments (learning rate, buffer size, bottleneck size). Authors argue that, building such theoretical models is important if the (neural) EC field wants to have a more general scientific impact on linguistics.

**Main review**

I really like author’s approach of developing a theoretical model. The choice of CRP makes sense as we expect the system to reinforce words that have a dominant frequency advantage in the system. I think the paper perfectly fits the topic of the workshop that aims to discuss the frontiers of (neural) EC. To me, the paper mainly asks the following question:

- To which extent can we propose theoretical models of emergent communication ? Can we link those models to properties observed with neural networks ?

For this reason, I recommend to accept the paper.

**Comments**

At this point of the research, I see no major weakness.

However, if the paper is a first draft for futur research, I would suggest to develop the following points:

- To me, the CRP process captures an important aspect of the reinforcement system: the more a word is used, the more it will be reused. However, the model lacks a core component of the system: the fact that there is an active listener with its own learning capacities. If a word has a high frequency but is not understood by the listener, the reward remains null: listener's understanding is required for frequency reinforcement. Recently [1] showed that the relative speaker-listener learning speed has actually a core impact on language properties including lexicon entropy. Indeed, it would be interesting to capture the role of the listener in the model and maybe add additional modeling elements describing listener's role.  This last comment introduces the question: what properties of language/words help listener's understanding ?

- Experiments (and entropy behavior) are very sensitive to other regularization factors such as entropy or KL regularization coefficients (when using REINFORCE) and the temperature coefficient for Gumbel-Softmax. I would have expected a study of the impact of the coefficient temperature in Gumbel-Softmax on the entropy and how it could be linked to parameters of FiLex.

- When reading the paper, I directly thought about some theoretical analysis that are made in evolutionary linguistics, as presented in [2] (ex: Replicator Dynamics [3]). I think it could be of interest to link this kind of theoretical analysis to previous attempts to model evolutionary dynamics. Moreover, do you have any idea of other self-reinforcing protocols that could fit language emergence settings ?

- *Concern*: I see this model as a powerful tool to analyze frequency phenomena of words. However, I have some difficulties to see how it could generalize to questions that are often asked in neural-EC: obtention of ZLA (eg.[4,5]), compositionality (eg. [6,7]), generalization (eg. [8,9]).  Adapting the FiLex to messages of variable lengths and to the analysis of those properties would be very nice.

**Details**

- Why using Gumbel-Softmax and not policy gradient theorem methods ?
- Do you report only results of successful experiments (ie. in term of performances) ? More generally, I think it would be interesting to add an analysis on the performances of the agents.
- Do you vary only speaker’s learning rate or speaker AND listener’s learning rates ?
- Which optimizer do you use ? SGD ? Adam ? If you use a second ordre method such as Adam, how relevant is the analogy between alpha parameter and the learning rate ?

**References**

- [1] Mathieu Rita, Florian Strub, Jean-Bastien Grill, Olivier Pietquin, Emmanuel Dupoux. On the role of population heterogeneity in emergent communication. International Conference on Learning Representations (ICLR) 2022
- [2] Brian Skyrms. Signals: Evolution, Learning, and Information (2010)
- [3] Maynard Smith, J.; Price, G.R. (1973). The logic of animal conflict. Nature. 246 (5427): 15–8.
- [4] Rahma Chaabouni, Eugene Kharitonov, Emmanuel Dupoux, and Marco Baroni. Anti-efficient encoding in emergent communication. In Proc. of Advances in Neural Information Processing Systems (NeurIPS), 2019.
- [5] Mathieu Rita, Rahma Chaabouni, and Emmanuel Dupoux. “LazImpa”: Lazy and impatient neural agents learn to communicate efficiently. In Proc. of the Conference on Computational Natural Language Learning (CoNLL), 2020.
- [6] Edward Choi, Angeliki Lazaridou, and Nando de Freitas. Multi-agent compositional communication learning
from raw visual input. In Proc. of International Conference on Learning Representations (ICLR), 2018.
- [7] Fushan Li and Michael Bowling. Ease-of-teaching and language structure from emergent communication. In
Proc. of Advances in Neural Information Processing Systems (NeurIPS), 2019.
- [8] Marco Baroni. Linguistic generalization and compositionality in modern artificial neural networks. Philosophical
Transactions of the Royal Society B: Biological Sciences, 375(1791):20190307, 2020. doi: 10.1098/rstb.2019.0307.
- [9] Rahma Chaabouni, Eugene Kharitonov, Diane Bouchacourt, Emmanuel Dupoux, and Marco Baroni. Compositionality
and generalization in emergent languages. In Proc. of the Association for Computational Linguistics (ACL), 2020.

---

### Official Review · Reviewer_tVP7 · 2022-03-24

**Rating:** Accept
**Confidence:** 4

**Review:**

### Summary
The paper proposes a theoretical model for emergent language systems, with the goal of creating a unified framework for understanding/explaining emergent communication protocols. The proposed framework extends the well-known Chinese restaurant process, introducing hyper parameters to account for the fixed-length nature of each agent’s lexicon and its use of a replay buffer during training. Empirical evaluation of the framework is performed in the context of a simple circular navigation task.

### Strengths
- Theoretical accounts of emergent behavior, and especially emergent communication, are an important open problem for RL systems and very relevant to this workshop. This work would be a great contribution.
- The paper is well-written and does a great job describing each aspect of the mathematical framework.
- The connection between self-reinforcing processes and patterns of word usage in natural languages is intuitive and well-motivated (a nice connection between language properties and theory).

### Suggestions
- My impression of this work is generally very positive. It may benefit, however, from a more formal introduction of CRP’s and mathematical/algorithmic discussion of FiLex
- The paper would also benefit from more detailed coverage of related work — there are a number of recent works in this area missing from the related works section.
    - If space is an issue, it seems that the “role of the model” paragraph in Section 3 (though a nice discussion), could be summarized in an introduction/conclusion.
- In addition to the circle navigation task, it would be nice/relevant to use this theoretical framework to study existing environments that have been introduced in prior works (navigation or otherwise).
    - See Lazaridou et al. 2020 for an overview of some of these works.

---

### Decision · Program_Chairs · 2022-03-25

**Decision:**

Accept

**Comment:**

This paper provides an interesting mathematical model for emergent communication and both reviewers find it intriguing and useful. We look forward to this discussion the novel direction of theoretical modelling of EC!